# Gapless spin liquid in a square-kagome lattice antiferromagnet

Masayoshi Fujihala [1]✉, Katsuhiro Morita[2]✉, Richard Mole[3], Setsuo Mitsuda[1], Takami Tohyama [2], Shin-ichiro Yano[4], Dehong Yu [3], Shigetoshi Sota[5], Tomohiko Kuwai[6], Akihiro Koda[7], Hirotaka Okabe[7], Hua Lee[7], Shinichi Itoh[8], Takafumi Hawai[8], Takatsugu Masuda [9], Hajime Sagayama[10], Akira Matsuo[11], Koichi Kindo[11], Seiko Ohira-Kawamura[12] & Kenji Nakajima [12]

Observation of a quantum spin liquid (QSL) state is one of the most important goals in condensed-matter physics, as well as the development of new spintronic devices that support next-generation industries. The QSL in two dimensional quantum spin systems is expected to be due to geometrical magnetic frustration, and thus a kagome-based lattice is the most probable playground for QSL. Here, we report the first experimental results of the QSL state on a square-kagome quantum antiferromagnet, $KCu_6AlBiO_4(SO_4)_5Cl$. Comprehensive experimental studies via magnetic susceptibility, magnetisation, heat capacity, muon spin relaxation ($\mu$SR), and inelastic neutron scattering (INS) measurements reveal the formation of a gapless QSL at very low temperatures close to the ground state. The QSL behavior cannot be explained fully by a frustrated Heisenberg model with nearest-neighbor exchange interactions, providing a theoretical challenge to unveil the nature of the QSL state.

[1] Tokyo University of Science, Department of Physics, Tokyo 162-8601, Japan. [2] Tokyo University of Science, Department of Applied Physics, Tokyo 125-8585, Japan. [3] Australian Nuclear Science and Technology Organisation, Lucas Heights, NSW 2232, Australia. [4] National Synchrotron Radiation Research Center, Hsinchu 30077, Taiwan. [5] Computational Materials Science Research Team, RIKEN Center for Computational Science, Kobe, Hyogo 650-0047, Japan. [6] Graduate School of Science and Engineering, University of Toyama, Toyama 930-8555, Japan. [7] Muon Science Laboratory and Condensed Matter Research Center, Institute of Materials Structure Science, High Energy Accelerator Research Organisation, 1-1 Oho, Tsukuba 305-0801, Japan. [8] Neutron Science Division, Institute of Materials Structure Science, High Energy Accelerator Research Organisation, 1-1 Oho, Tsukuba, Ibaraki 305-0801, Japan. [9] Institute for Solid State Physics, The University of Tokyo, Kashiwa, Chiba 277-8581, Japan. [10] Synchrotron Radiation Science Division 1 and Center for Integrative Quantum Beam Science, Institute of Materials Structure Science, High Energy Accelerator Research Organization, 1-1 Oho, Tsukuba, Ibaraki 305-0801, Japan. [11] International MegaGauss Science Laboratory, Institute for Solid State Physics, The University of Tokyo, Kashiwa, Chiba 277-8581, Japan. [12] Materials and Life Science Division, J-PARC Center, Tokai, Ibaraki 319-1195, Japan. ✉email: fujihara@nsmsmac4.ph.kagu.tus.ac.jp; katsuhiro.morita@rs.tus.ac.jp

Magnetic phases of low-dimensional magnets have been studied both theoretically and experimentally in the last half century. Intensive studies of one-dimensional (1D) spin systems have successfully captured the exotic quantum states, such as the Tomonaga–Luttinger spin-liquid state[1] and the Haldane state[2]. Recent progress in synthesising ideal 1D magnets has evolved this research field[3]. On the other hand, in 2D spin systems, the spin-1/2 kagome antiferromagnet is an excellent choice for searching for the QSL state induced by geometrical frustration[4]. A possible compound for QSL in the kagome antiferromagnets was herbertsmithite, which has the $Cu^{2+}$ layers with ideal kagome geometry sandwiched by nonmagnetic $Zn^{2+}$ layers[5]. To date, no long-range order has been found at any temperature, and a continuum of spin excitations was observed by INS experiments. However, the low-energy magnetic excitation is still unclear as seen in a controversy on gapless[6] or gapped[7] excitation. This is related to the fact that herbertsmithite is obtained by selectively replacing magnetic $Cu^{2+}$ ions with nonmagnetic $Zn^{2+}$ ions on the triangular-lattice planes of its parent compound clinoatacamite[8], $Cu_2(OH)_3Cl$. This replacement inevitably causes site mixing[9]. Other materials with the kagome lattice exhibit long-range magnetic or valence-bond crystal (VBC) orders caused by lattice distortions, the DM interaction and further neighbour interactions[10–14]. The lack of a suitable model material exhibiting the QSL hinders observations of the QSL state in the 2D spin-1/2 systems.

Another highly frustrated 2D quantum spin system expected to be a QSL state is a compound with the square-kagome lattice (SKL). The SKL was introduced by Siddharthan et al.[15]. It has the same coordination number as the kagome lattice ($z = 4$), but with a composition of two inequivalent sublattices in contrast to the kagome lattice. Richter et al. reported that the ground state of the spin-1/2 SKL with three equivalent exchange interactions (the case of $J_1 = J_2 = J_3$ and $J_X = 0$ in Fig. 1c) is similar to that of the kagome lattice[16]. The ground state of the spin-1/2 $J_1$–$J_2$ SKL antiferromagnet (the case of $J_2 = J_3$ and $J_X = 0$ in Fig. 1c) was calculated by Rousochatzakis et al.[17]. It has been predicted to be a crossed-dimer VBC state and a square pinwheels VBC state, depending on $J_2/J_1$. Moreover, there is a possibility that the QSL ground states are realised in the SKL with three nonequivalent exchange interactions (the case of $J_X = 0$ in Fig. 1c), which lead to the melting of these VBC states[18]. Very recently, it has also

predicted to be a topological nematic spin-liquid state[19]. In the magnetic field, the existence of the magnetisation plateaus of $M/M_{sat} = 1/3$ and $2/3$ has theoretically clarified[16–18,20], where $M_{sat}$ is the saturation magnetisation. These plateau phases exhibit VBC, up–up–down structure, and alternate trimerized states. In the high magnetic field and low-temperature regime, a magnetic-field-driven Berezinskii–Kosterlitz–Thouless phase transition exists[21]. However, the lack of a model compound for the SKL system has obstructed a deeper understanding of its spin state. Motivated by the present status on the study of the SKL system, we searched for compounds with the SKL containing $Cu^{2+}$ spins, and synthesised the first compound of a SKL antiferromagnet, $KCu_6AlBiO_4(SO_4)_5Cl$, successfully. Here, we use thermodynamic, muon spin relaxation and neutron-scattering experiments on powder samples of $KCu_6AlBiO_4(SO_4)_5Cl$, to demonstrate the absence of magnetic ordering and the presence of gapless continuum of spin excitations.

## Results

**Crystal structure.** The synthesis of $KCu_6AlBiO_4(SO_4)_5Cl$ was conceived following the identification of the naturally occurring mineral atlasovite, $KCu_6FeBiO_4(SO_4)_5Cl[22]$. The space group and structural parameters of $KCu_6AlBiO_4(SO_4)_5Cl$ are determined as $P4/ncc$, (the same space group as atlasovite) and $a = 9.8248(9)$ Å, $c = 20.5715(24)$ Å, respectively (see Supplementary Note 1). As shown Fig. 1a and b, the SKL in the crystal structure of $KCu_6AlBiO_4(SO_4)_5Cl$ comprises the six-coordinated $Cu^{2+}$ ions. In each SK unit, the square is enclosed by four scalene triangles. From this crystal structure, it is recognised that $KCu_6AlBiO_4(SO_4)_5Cl$ has three types of first neighbour interactions, $J_1$, $J_2$ and $J_3$, as shown in Fig. 1c. The orbital arrangements can be reasonably deduced from the oxygen and chloride positions around the $Cu^{2+}$ ions. Judging from the $d_{x^2-y^2}$ orbitals arranged on the SKL, the nearest-neighbour (NN) magnetic couplings $J_i$ ($i = 1$–3) are superexchange interactions occurring through Cu–O–Cu bonds: $J_1$ through the Cu1–O–Cu1 bond with a bond of angle 112.62°, and $J_2$ and $J_3$ through Cu1–O–Cu2 with bond angles of 120.12° and 108.61°, respectively. Since the Cu–O–Cu angle significantly influences on the value of the exchange interactions, the variation of the angles can give strong bond-dependent exchange interactions[23]. Therefore, $J_2$ with the largest angle is expected to

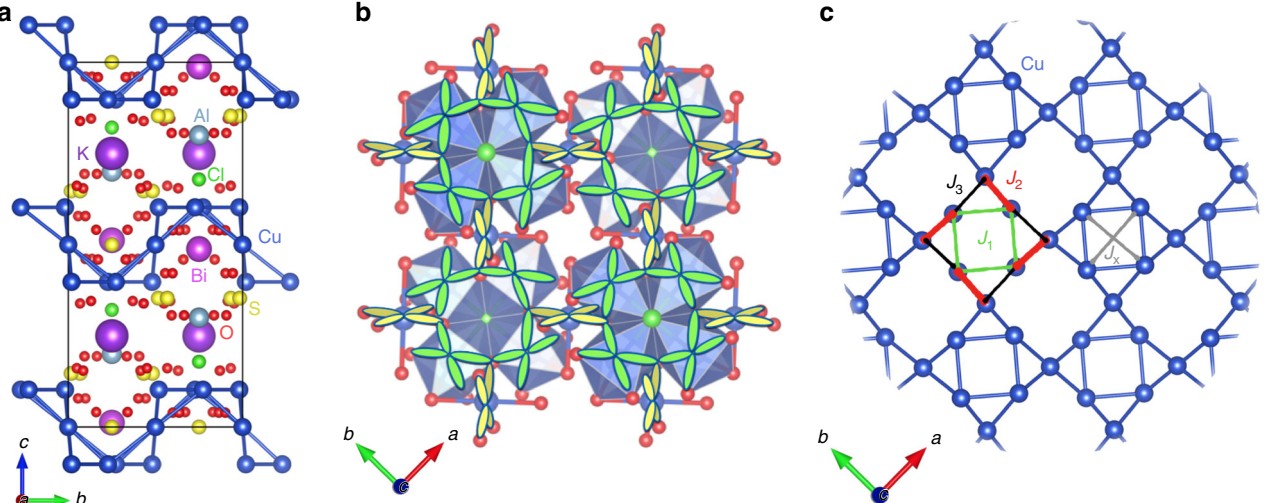

**Fig. 1 Spin-1/2 $J_1$–$J_2$–$J_3$ square-kagome lattice in $KCu_6AlBiO_4(SO_4)_5Cl$. a** Crystal structure of $KCu_6AlBiO_4(SO_4)_5Cl$ featuring a large interlayer spacing. **b** Arrangement of the $Cu^{2+}$ orbitals in SKL. The $d_{x^2-y^2}$ orbitals carrying spin-1/2 are depicted on the Cu sites. **c** Square-kagome lattice of $KCu_6AlBiO_4(SO_4)_5Cl$ consisting of $Cu^{2+}$ ions with nearest-neighbour exchange couplings $J_1$, $J_2$, $J_3$ and next-nearest-neighbour exchange coupling $J_X$.

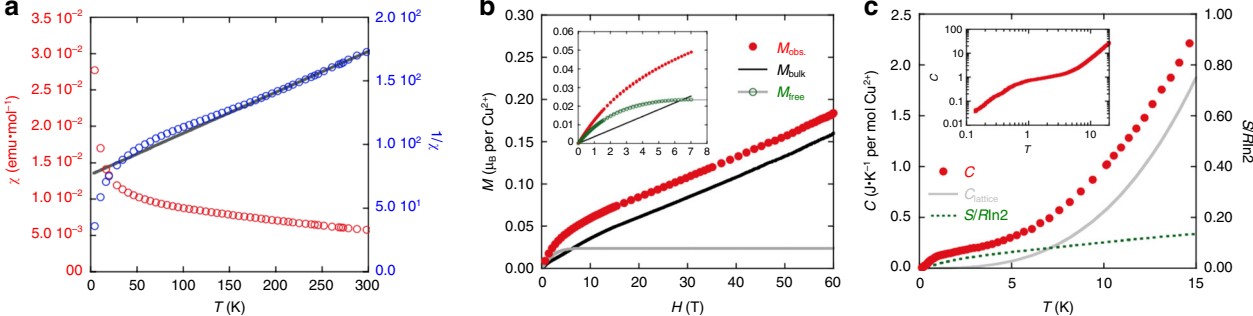

**Fig. 2 Magnetic and thermodynamic properties of KCu$_6$AlBiO$_4$(SO$_4$)$_5$Cl. a** Temperature dependence of the magnetic susceptibility $\chi$ (open red circles) and the inverse susceptibilities $1/\chi$ (open blue circles) of KCu$_6$AlBiO$_4$(SO$_4$)$_5$Cl measured at 1 T. The $\chi$ is obtained by subtracting the Pascal's diamagnetic contribution from the experimental data. The solid grey lines denote the fitting curves by the Curie–Weiss law. **b** High-field magnetisation measured up to 60 T at 1.8 K. The observed data $M_{obs.}$ (filled red circles) are broken down into two components: $M_{bulk}$ (black solid line) and $M_{free}$ (open green circles). Inset shows the magnetisation measured using MPMS at 1.8 K. The grey line is the Brillouin function for $g = 2$ and 2.4% of free $S = 1/2$ spins. **c** Temperature dependence of the total specific heat measured at zero field (filled red circles). The grey line is the assumed lattice contribution $C_{lattice.} = 0.000555T^3$. The green dashed line is the estimated magnetic entropy. Inset shows a log-log plot of the same data.

be the largest antiferromagnetic interaction, while $J_3$ with the smallest angle is considered to be the smallest antiferromagnetic interaction among the three interactions. One prominent and important feature of the present structure is the occupancy of nonmagnetic atoms in the interlayer space of the unit cell (Fig. 1b), which elongate the interlayer spacing. Furthermore, the Cu$^{2+}$ ions and nonmagnetic ions have different valence numbers in KCu$_6$AlBiO$_4$(SO$_4$)$_5$Cl, avoiding site mixing, unlike the Cu$^{2+}$ and Zn$^{2+}$ site mixing observed in herbertsmithite (for more details, see Supplementary Notes 1 and 2). Therefore, the crystal perfectness and high two-dimensionality of KCu$_6$AlBiO$_4$(SO$_4$)$_5$Cl are ideal for studying the intrinsic magnetism on frustrated 2D magnets. However, the obtained INS experimental results are inconsistent with the calculated results for the $J_1$-$J_2$-$J_3$ SKL model (discussed below).

**Magnetic and thermodynamic properties.** Figure 2a presents the temperature dependence of the magnetic susceptibility $\chi(T)$ and the inverse magnetic susceptibility $1/\chi(T)$ of KCu$_6$AlBiO$_4$(SO$_4$)$_5$Cl in the temperature range 1.8–300 K. With decreasing temperature, the magnetic susceptibility gradually increases. This feature suggests the absence of any long-range order down to 1.8 K. From $1/\chi(T)$ with the Curie–Weiss law $C/(T-\theta_{CW})$, between 200 K and 300 K, we estimated the Curie constant and Weiss temperature to be $C = 2.86(1)$ and $\theta_{CW} = -237(2)$ K, respectively. The $C$ corresponds to an effective moment of $1.96\,\mu_B$, consistent with the spin $S = 1/2$ of Cu$^{2+}$. The large negative Weiss temperature and the absence of long-range orders suggest an antiferromagnetic frustrated system.

The magnetisation curve measured at 1.8 K, as shown in the inset of Fig. 2b, has two components: an intrinsic component $M_{bulk}$ and free spin component $M_{free}$. Following the analysis for herbertsmithite[24], a saturated magnetisation of $M_{free}$ can be estimated by subtraction of the linear $M_{bulk}$ from the measured total magnetisation $M_{obs.}$. The $M_{free}$ can be fitted a Brillouin function for a spin-1/2, suggesting the component is attributed to the paramagnetic impurity or the unpaired spins on surface of powder particles. The saturated value of $M_{free}$ indicates the presence of free spins with about 2.4% in total Cu$^{2+}$ ions in our sample. The $M_{bulk}$ at high magnetic field is only ~0.15 $\mu_B$/Cu$^{2+}$ at 60 T, indicating that strong antiferromagnetic exchange interaction dominates in this system (see Fig. 2b).

A Schottky-like peak in the heat capacity is observed at around $T^* \approx 2$ K, as shown in Fig. 2c. As the released magnetic entropy at 15 K is only 16% of the expected total entropy, which is similar to

that of herbertsmithite. In herbertsmithite, this behaviour was attributed to weakly coupled spins residing on the interlayer sites[9]. However, in KCu$_6$AlBiO$_4$(SO$_4$)$_5$Cl, it is difficult to assign the 16% entropy to the site mixing because of the valence of nonmagnetic ions different from Cu$^{2+}$. Rather the observed peak can be attributed easily to the development of short-range spin correlations. Similar characteristics are observed in the calculated specific heat and entropy of the the spin-1/2 kagome antiferromagnet[25]. Small broad peak appear at around $T \approx J/100$, and the released entropy at around this temperature is about 20%. As discussed below, the magnitude of the exchange interaction $J_{av} \equiv (J_1 + J_2 + J_3)/3 = 137$ K for KCu$_6$AlBiO$_4$(SO$_4$)$_5$Cl, namely, $J_{av}/100 \approx T^*$. However, careful consideration is necessary about what origin of this peak is. We therefore conclude that the long-range magnetic and VBC-ordering behaviours are not observed in magnetic susceptibility, magnetisation and specific heat.

**Quantum spin fluctuations in KCu$_6$AlBiO$_4$(SO$_4$)$_5$Cl.** To confirm the absence of spin ordering caused by quantum fluctuations, we performed $\mu$SR measurements. Figure 3a shows the weak longitudinal-field (LF) (=50 G) $\mu$SR spectra at various temperatures. The weak LF was applied to quench the depolarisation due to random local fields from nuclear magnetic moments. The spectra are well fitted by the exponential function

$$a(t) = a_1 \exp(-\lambda t) + a_{BG}, \qquad (1)$$

where $a_1$ is an intrinsic initial asymmetry $a_1 = 0.133$, $a_{BG}$ is a constant background $a_{BG} = 0.047$ (see Supplementary Note 2), $\lambda$ is the muon spin relaxation rate. Hartree potential calculation predicted a local potential minimum in the lattice (see Fig. 3d, e)[26–28]. A muon site corresponding to a local potential minimum is located at the 16g site. Quantum fluctuations of the Cu$^{2+}$ spins down to 58 mK without spin ordering/freezing are evidenced by the long-time $\mu$SR spectra. The weak LF signals at the lowest temperature (58 mK) decrease continuously without oscillations up to 15 $\mu$s, as shown in Fig. 3b. If this spectrum is due to static magnetism, the internal field (estimate as $\lambda_{ZF}/\gamma_\mu$, where $\gamma_\mu$ is the muon gyromagnetic ratio) should be less than 20 G. (see Supplementary Note 3). However, relaxation is clearly observed in the LF spectrum even at 0.395 T, which is evidence for the fluctuation of Cu$^{2+}$ electron spins without spin ordering/freezing (see Fig. 3c). As shown in Fig. 3f, the increase of $\lambda$ at around $T^*$ renders evidence for a slowing down of the spin fluctuation resulting from the development of short-range correlations. In addition, they exhibit a plateau with weak temperature

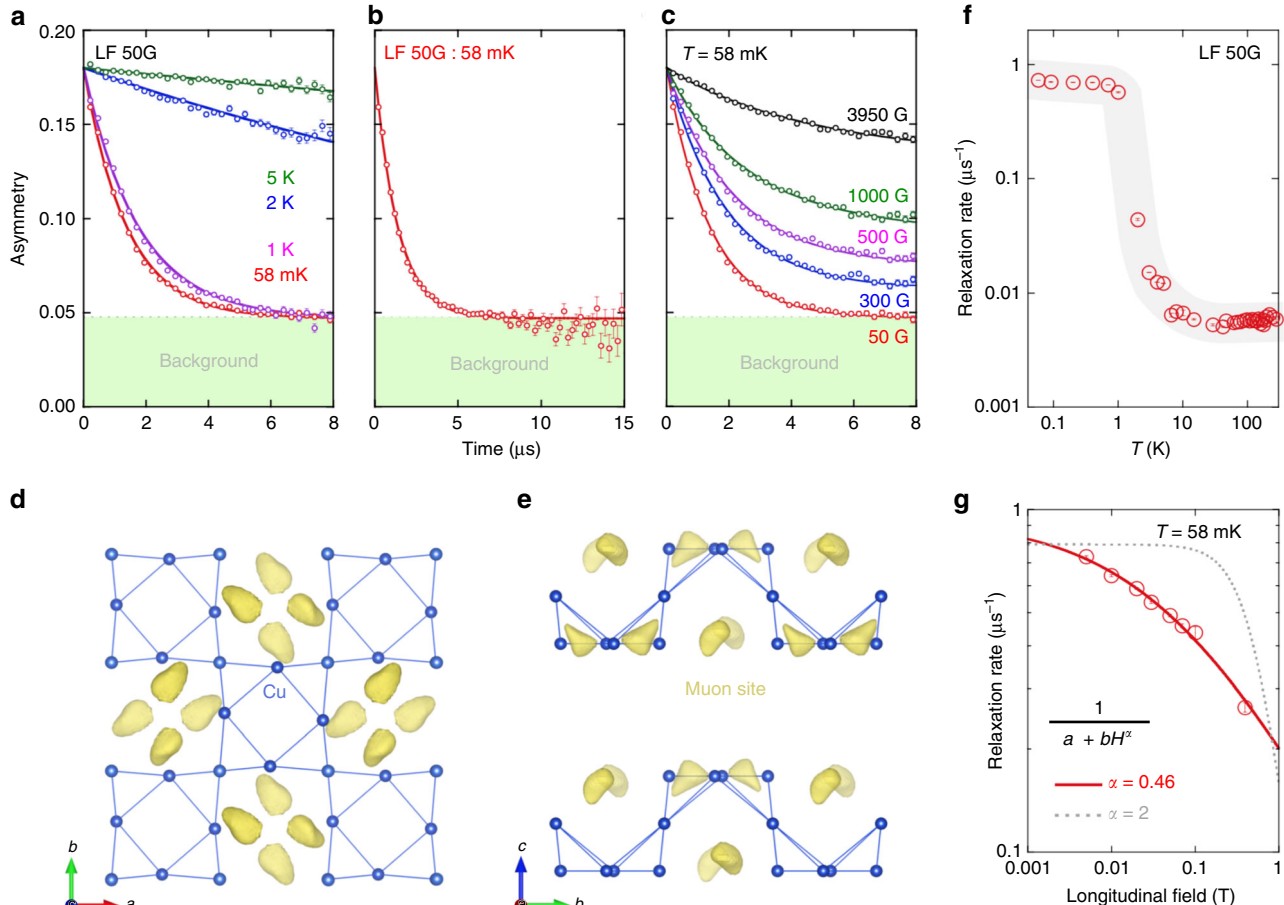

**Fig. 3 Muon spin relaxation data of KCu$_6$AlBiO$_4$(SO$_4$)$_5$Cl. a** LF-$\mu$SR spectra (obtained in a dilution refrigerator) at representative temperatures (see Supplementary Note 3 for the spectra obtained using the $^4$He cryostat). The thick lines behind the data points are the fitted curves (see text for details). **b** The LF-$\mu$SR spectrum measured at 58 mK. The spectrum decreases continuously without oscillations up to 15 $\mu$s. **c** $\mu$SR spectra measured at 58 mK under several longitudinal magnetic fields. **d** Projection along the $c$ axis. **e** Projection along the $a$ axis. The muon site was obtained by a Hartree potential calculation. **f** Temperature dependence of the muon spin relaxation rate $\lambda$. The grey solid lines are guides for the eyes. **g** Magnetic-field dependences of the muon spin relaxation rate $\lambda$. The solid curves are fitted to a power law of the form $1/(a + bH^\alpha)$. The error bars in **a**, **b** and **c** represent 1 s.d. and in **f** and **g** the maximum possible variation due to correlation of parameters.

dependence at low temperature, which has been found in other QSL candidates[29]. The LF spectra measured at 58 mK under several magnetic fields are also fitted by Eq. (1). Using the power law represented by $1/(a + bH^\alpha)$ with $\alpha = 0.46$, where $a$ and $b$ depend on the fluctuation rate and fluctuating field, we obtain a good fitting to the LF dependence of the muon spin relaxation rate $\lambda$, as shown in Fig. 3g. Incidentally, the $1/(a + bH^2)$ is a standard case that the $\lambda$ obeys the Redfield equation. In ordinary disordered spin systems, the muon spin relaxation rate exhibits a field- inverse square dependence. Such a spectral-weight function is commonly used to describe classical fluctuations in the paramagnetic regime. The observed values, $\alpha = 0.46$, are inconsistent with the existence of a single timescale and suggest a more exotic spectral density, such as the one at play in a QSL. All of these $\mu$SR results strongly support the formation of a QSL at very low temperature close to the ground state in KCu$_6$AlBiO$_4$(SO$_4$)$_5$Cl[30,31].

**Gapless continuum of spin excitations**. The quantum statistics of quasiparticle excitations depend on the type of QSL, in particular, the nature of their excitation. To grasp the whole picture of the spin excitation, first we performed the INS experiment in a wide energy range. As shown in Fig. 4a, streak-like excitation at

$Q = 0.8$ Å$^{-1}$ and flat signals at around $E = 7$ and 10 meV are observed at 5 K. The $E$-dependence of the INS intensity can be fitted well by two or three Gaussian functions and linear baseline, and the corresponding integrated intensities are obtained (for more details, see Supplementary Note 4). As shown in Fig. 4b, the peak positions of excitations are estimated to be 10.1(1) meV, 9.4 (3) meV, and 7.3(1) meV, respectively. The signal due to magnetic excitation is generally enhanced at low-$Q$ values, whereas phonon excitation is dominant at high-$Q$. As shown in Fig. 4c, the baseline increase with increasing with $Q$. Therefore, the baseline may well comes from a number of phonon excitations in a multi-element material KCu$_6$AlBiO$_4$(SO$_4$)$_5$Cl. The peak at 9.4 meV also increases with increasing with $Q$, indicating that it comes from phonon excitation. On the other hand, the flat signals have a characteristic feature of magnetic excitation. In order to investigate whether the spin excitation is gapless or gapped, we performed the INS experiments in the low-energy region. These signals are also observed at 0.3 K, as shown in Fig. 4d, there are the streak-like excitation and flat signals are also observed. As shown in Fig. 4e and g, the INS spectra exhibit the feature of a gapless continuum of spin excitations. Streak-like excitation at $Q = 0.8$ Å$^{-1}$ is clearly visible down to the elastic line, and its intensity increases continuously without signature of energy gap at least within the instrumental resolution (FWHM = 0.05 meV

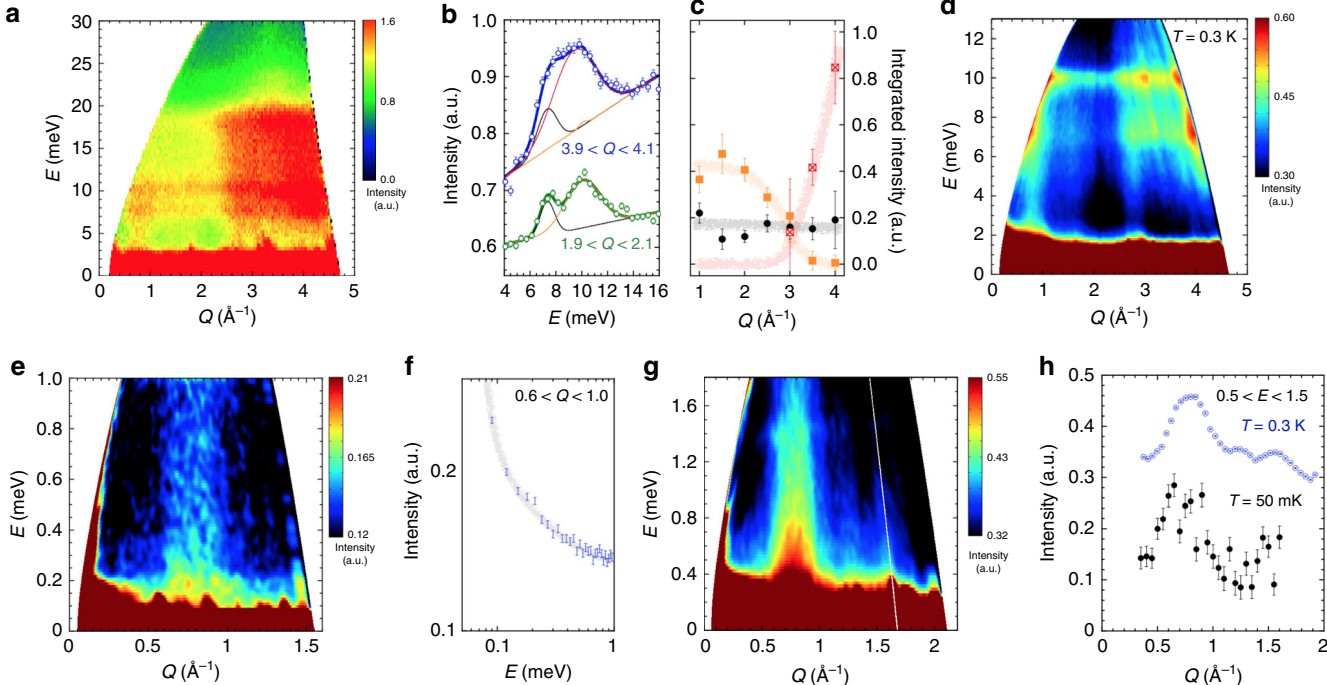

**Fig. 4 Inelastic neutron-scattering data of KCu$_6$AlBiO$_4$(SO$_4$)$_5$Cl. a** INS spectra at 5 K observed using HRC with an incident neutron energy of 45.95 meV. **b** Energy dependence of the scattering integrated over $Q$ in the range 1.9 Å$^{-1}$ < $Q$ < 2.1 Å$^{-1}$ and 3.9 Å$^{-1}$ < $Q$ < 4.1 Å$^{-1}$ measured at 5 K (HRC). The solid lines are the fitted curves (see text for details), the thin lines are its components. **c** $Q$-dependence of the integrated intensity for the different Gaussian components ($E$ = 10.1(1) meV, 9.4(3) meV and 7.3(1) meV). The solid thick lines are guides for the eyes. **d** INS spectra at 0.3 K observed using AMATERAS with an incident neutron energy of 15.16 meV. **e** INS spectra at 0.3 K observed using AMATERAS with incident neutron energy of 1.69 meV. **f** Energy dependence of the scattering integrated over $Q$ in the range 0.6 Å$^{-1}$ < $Q$ < 1.0 Å$^{-1}$ measured at 0.3 K. The grey solid line is guides for the eyes. **g** INS spectra at 0.3 K observed using AMATERAS with an incident neutron energy of 3.14 meV. **h** $Q$-dependence of the scattering integrated over energy transfers 0.5 meV < $E$ < 1.5 meV measured at 0.3 K (AMATERAS) and 50 mK (PELICAN). The error bars represent standard deviation.

for $E_i$ = 1.69 meV). The excitation persist up to at least $T$ = 30 K (see Supplementary Fig. 5), which is consistent with the exchange constants estimated later. The $Q$-dependence of the INS intensity after integration over a finite energy interval is shown in Fig. 4h. There are three peaks at $Q$ = 0.8, 1.25, and 1.58 Å$^{-1}$ at 0.3 K, and the peaks are observed even at low temperatures close to the ground state (48 mK). As discussed below, this result is inconsistent with the calculated dynamical spin structure factor $S(\mathbf{q}, \omega)$ in the $J_1$–$J_2$–$J_3$ SKL antiferromagnet with parameters, which reproduce magnetic susceptibility and magnetisation process. These INS data are consistent with a gapless continuum of spinon excitations. From the above, the flat signals at approximately 10 and 7 meV probably indicate a van Hove singularity of spinon continuum edges at this energy.

**Comparison with theory.** To determine the magnetic parameters and to clarify the magnetic properties of KCu$_6$AlBiO$_4$(SO$_4$)$_5$Cl, we calculated the magnetic susceptibility, the magnetisation curve at zero temperature, and the magnetic excitation at zero temperature by mean of the exact diagonalization (ED), finite temperature Lanczos (FTL)[32] and density-matrix renormalization group (DMRG) method. We succeeded in reproducing the magnetic susceptibility and magnetisation curve of KCu$_6$Al-BiO$_4$(SO$_4$)$_5$Cl with the $J_1$–$J_2$–$J_3$ SKL model with $J_1$ = 135 K, $J_2$ = 162 K, $J_3$ = 115 K and $g$ = 2.11, as shown in Fig. 5a and b, where $g$ is the gyromagnetic ratio. In the magnetisation process, the magnetisation plateaus of $M/M_{sat}$ = 1/3 and 2/3 were confirmed at around 150 T and 270 T, respectively. This indicates the possibility to observe magnetisation plateaus experimentally if the measurement of the magnetisation process in a further strong magnetic field is performed. However, the result of inelastic

neutron scattering that is the most important evidence of QSL cannot be reproduced in the $J_1$–$J_2$–$J_3$ SKL with these parameters. In the inelastic neutron-scattering experiment, in the low-energy region, the strongest intensity become around $Q$ = 0.8 Å$^{-1}$ as shown Fig. 4f, g, while in the dynamical DMRG method, it is around $Q$ = 1.3 Å$^{-1}$ as shown in Fig. 5c. To eliminate this discrepancy, we also calculated the SKL model with next-nearest-neighbour (NNN) interaction $J_X$ in the diagonal direction of the Cu$^{2+}$ square. We calculated this SKL model with various values of the parameters, but we could not reproduce the experimental results. Therefore, in order to understand the experiment correctly, we need to calculate the model with further interactions.

**Discussion**

We have synthesised a SKL spin-1/2 antiferromagnet KCu$_6$Al-BiO$_4$(SO$_4$)$_5$Cl without site disorder, thus providing a first candidate to investigate the SKL magnetism. The $\mu$SR measurement shows no long-range ordering down to 58 mK, roughly three orders of magnitude lower than the NN interactions. The INS spectrum exhibits a streak-like gapless excitation and flat dispersionless excitation, consistent with powder-averaged spinon excitations. Our experimental results strongly suggest the formation of a gapless QSL in KCu$_6$AlBiO$_4$(SO$_4$)$_5$Cl at very low temperature close to the ground state; however, they are inconsistent with the theoretical studies based on the $J_1$–$J_2$–$J_3$ SKL Heisenberg model. In the $J_1$–$J_2$–$J_3$ SKL Heisenberg model, the VBC and Néel order stats are expected with high probability. In fact, the VBC state is the ground state of the $J_1$–$J_2$ SKL antiferromagnet regardless of the magnitude relation of $J_1/J_2$[17]. Thus, to realise the QSL state in the SKL, we must impose an additional condition such as longer-range exchange interactions. Further

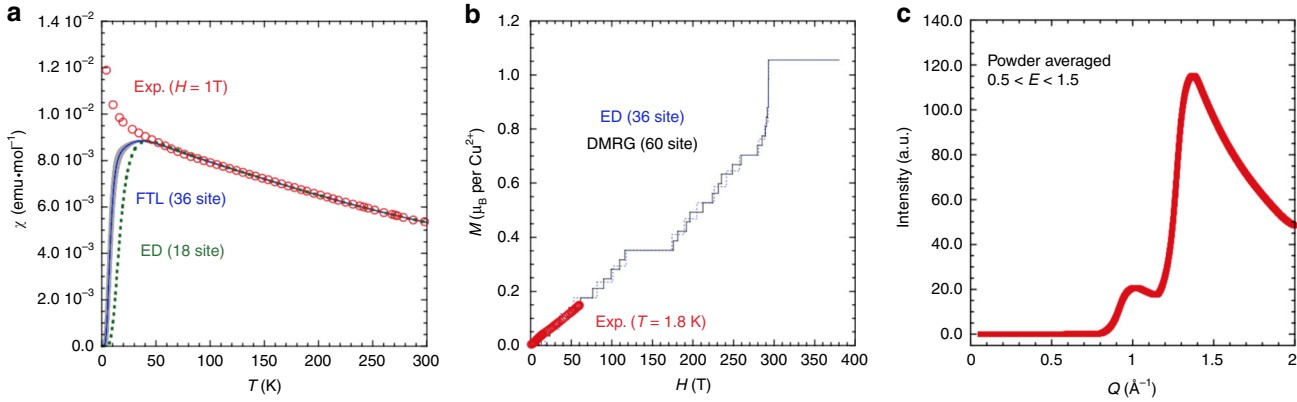

**Fig. 5 Experimental results of KCu₆AlBiO₄(SO₄)₅Cl compared to theory. a** Temperature dependence of the magnetic susceptibility $\chi$ (open red circles) of KCu₆AlBiO₄(SO₄)₅Cl and the fitted calculation data obtained by the FTL method for a 36-site cluster (blue line) and ED method for a 18-site cluster (green dashed line). Note that the statistical error of the FTL is within the grey area (for more details, see Supplementary Note 5). **b** High-field magnetisation measured up to 60 T at 1.8 K (open red circles) and the fitted calculation data at $T = 0$ K obtained by the Lanczos-type ED method for a 36-site cluster (blue dashed line) and DMRG method for a 60-site cluster (black solid line). **c** $Q$-dependence of powder-averaged dynamical spin structure factor $S(Q, E)$ integrated over 0.5 meV $< E < 1.5$ meV at $T = 0$ K obtained by dynamical DMRG for a 48-site PBC cluster of the SKL.

theoretical study would reveal the conditions inducing the QSL state in SKL antiferromagnets.

## Methods

**Sample synthesis**. Single phase polycrystalline KCu₆AlBiO₄(SO₄)₅Cl was synthesised by the solid-state reaction in which high-purity KAl(SO₄)₂, CuCl₂, CuSO₄, CuO and Bi₂O₃ powders were mixed in a molar ratio of 2:1:6:5:1, followed by heating at 600 °C for 3 days and slow cooling in air.

**X-ray diffraction**. Synchrotron powder XRD data were collected using an imaging plate diffractometer installed at the BL-8B of the Photon Factory. The incident synchrotron X-ray energy of 18.0 keV (0.68892 Å) was selected.

**Magnetic susceptibility and low-field magnetisation**. Magnetic susceptibility and low-field magnetisation measurements were performed using a commercial superconducting quantum interference device magnetometer (MPMS-XL7AC: Quantum Design).

**High-field magnetisation**. High-field magnetisation measurements up to 60 T were conducted using an induction method in a pulsed magnetic field at the International MegaGauss Science Laboratory, The University of Tokyo.

**Heat capacity**. The specific heat was measured between 0.2 and 20 K using a PPMS (physical property measurement system; Quantum Design).

**Muon spin relaxation ($\mu$SR)**. The $\mu$SR experiments were performed using the spin-polarised pulsed surface-muon ($\mu^+$) beam at the D1 beamline of the Materials and Life Science Experimental Facility (MLF) of the Japan Proton Accelerator Research Complex (J-PARC). The spectra were collected in the temperature range from 58 mK to 300 K using a dilution refrigerator and $^4$He cryostat.

**Inelastic neutron scattering (INS)**. The high-energy INS experiment was performed on the HRC[33], installed at BL12 beamline at MLF of J-PARC. At the HRC, white neutrons are monochromatised by a Fermi chopper synchronised with the production timing of the pulsed neutrons. The energy transfer was determined from the time-of-flight of the scattered neutrons detected at position sensitive detectors. A 200-Hz Fermi chopper was used to obtain a high neutron flux. A GM-type closed cycle cryostat was used to achieve 5 K. The energy of incident neutrons were $E_i = 45.95$ meV. The data collected by HRC were analysed using the software suite HANA[34]. The low-energy INS experiments were performed using the cold-neutron time-of-flight spectrometer PELICAN at the OPAL reactor at ANSTO[35]. The instrument was aligned for an incident energy $E_i = 2.1$ meV. The sample was held in an oxygen-free copper can and cooled using a dilution insert installed in a top-loading cryostat and data collected at 25 K, 15 K and 48 mK. The sample was corrected for background scattering from an empty can and normalised to the scattering from a vanadium standard. The PELICAN data corrections were performed using the freely available LAMP software. The INS spectra in a wide momentum-energy range were measured using the cold-neutron disk chopper spectrometer AMATERAS installed in the MLF at J-PARC[36]. The sample was cooled to 0.3 K using a $^3$He refrigerator. The

scattering data were collected with a set of incident neutron energies, $E_i = 1.69$, 3.14 and 15.16 meV. The data collected by AMATERAS were analysed using the software suite UTSUSEMI[37].

**Calculations**. Magnetic susceptibility of the SKL is calculated by the full ED method for 18-site and FTL method for 36-site under the periodic boundary condition (PBC). The result of the FTL method is deduced by the statistical average of 40 sampling. The magnetisation curve at $T = 0$ K is calculated by the Lanczos-type ED calculations for a 36-site PBC cluster and the DMRG method for a 60-site PBC cluster. The truncation number in the DMRG calculation is 6000 and resulting truncation errors are less than $2 \times 10^{-5}$. The dynamical spin structure factor $S(\mathbf{q}, \omega)$ is calculated using the dynamical DMRG[38] method for a 48-site PBC cluster. The truncation number $m = 6000$ and the truncation error are less than $5 \times 10^{-3}$. $S(\mathbf{q}, \omega)$ is defined as follows:

$$S(\mathbf{q}, \omega) = -\frac{1}{\pi N}\text{Im}\langle 0|S^z_{-\mathbf{q}}\frac{1}{\omega - \mathscr{H} + E_0 + i\eta}S^z_{\mathbf{q}}|0\rangle, \qquad (2)$$

where $\mathbf{q}$ is the momentum, $|0\rangle$ is the ground state with energy $E_0$, $\eta$ is a broadening factor and $S^z_{\mathbf{q}} = N^{-1/2}\sum_i e^{i\mathbf{q}\mathbf{r}_i}S^z_i$ with $\mathbf{r}_i$ being the position of spin $i$ and $S^z_i$ being the $z$ component of $\mathbf{S}_i$. The value of $\eta$ is taken to be 1.16 meV.

## Data availability
The data that support the findings of this study are available from the corresponding author upon reasonable request.

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

## Acknowledgements

The μSR and INS experiments were performed at the MLF of J-PARC under a user programme (Proposal Nos. 2017B0019, 2017B0039, 2017B0049 and 2018B0068). Synchrotron powder XRD measurements were performed with the approval of the Photon Factory Program Advisory Committee (Proposal No. 2016G030). Theoretical study is in part supported by Creation of new functional devices and high-performance materials to support next-generation industries (GCDMSI) to be tackled by using post-K computer and by MEXT HPCI Strategic Programs for Innovative Research (SPIRE) (hp160222, hp170274). This study is partly supported by the Grant-in-Aid for Scientific Research (No. 17K14344) from MEXT, Japan.

## Author contributions

M.F. and K.M. conceived the study. H.S. and M.F. performed the synchrotron XRD experiment. T.K. performed the specific heat measurement. A.M. and K.K. performed the high-field magnetisation measurement. A.K., H.O., H.L. and M.F. performed the μSR experiments. K.M. and T.T. performed numerical calculations by ED and DMRG developed by S.S. R.M., D.Y., S.Y., S.I., T.H., T.M., S.O.K., K.N., M.F. and S.M. performed the neutron-scattering experiments. All the authors contributed to the writing of the paper.

## Competing interests

The authors declare no competing interests.
