## [Peer Review File · Nature Communications]

Reviewers' comments:

Reviewer #1 (Remarks to the Author):

In this paper, the authors report about the first compound, $\text{KCu}_6\text{AlBiO}_4(\text{SO}_4)_5\text{Cl}$, which is of relevance to the square-kagome-lattice quantum Heisenberg antiferromagnet. They synthesized $\text{KCu}_6\text{AlBiO}_4(\text{SO}_4)_5\text{Cl}$, determined its crystal structure, and then report a set of experimental results supported by theoretical calculations to check whether a gapless quantum spin liquid state is realized in this compound as $T \rightarrow 0$. More specifically, they measured the zero-field magnetic susceptibility $\chi(T)$, the magnetization curve $M(H)$, the zero-field specific heat $C(T)$, and performed the muon spin relaxation and inelastic neutron scattering measurements to provide evidence for a gapless quantum spin liquid state at $T = 0$ in this material. From theoretical side, the authors estimated the exchange couplings for the square-kagome-lattice quantum Heisenberg antiferromagnet, $J_1 = 135$ K, $J_2 = 162$ K, and $J_3 = 115$ K, and the g-factor, $g = 2.11$, which can reproduce the experimental data for magnetic susceptibility and magnetization curve, although fail to explain the inelastic neutron scattering experiment.

In my opinion, this is an interesting paper which deserves to be published in Nature Communications. Search for solid-state realizations of a quantum spin liquid state is rather hot topic nowadays and the communication about a real-life square-kagome-lattice $S = 1/2$ Heisenberg antiferromagnet sounds very exciting. The reported results are novel and, in my opinion, they may influence thinking in quantum magnetism community and inspire new theoretical and experimental studies.

However, I have to advise the authors to consider the following points.

- First of all, they must mention somehow that theoreticians studied this model at least since 2002, i.e., before the paper by I. Rousochatzakis et al. (2013) [19]. I mean the following important papers: R. Siddharthan and A. Georges, Square kagome quantum antiferromagnet and the eight-vertex model, Phys. Rev. B 65, 014417 (2002) and J. Richter, J. Schulenburg, P. Tomczak, and D. Schmalfuß, The Heisenberg antiferromagnet on the square-kagome lattice, Condensed Matter Physics 12, 507 (2009); unfortunately, both papers are missed. By the way, in the latter paper one can find the ground-state magnetization curve as it follows from ED ($N = 24 \dots 54$) for the $J_1 = J_2 = J_3$ case (Fig. 5), which looks similar to what is shown in Fig. 5(b) for slightly different values of J_1 , J_2 , and J_3 ($J_2 = 1.2J_1$, $J_3 = 0.85J_1$). Furthermore, the case $J_2 \neq J_3$ was studied in Phys. Rev. B 88, 094426 (2013). The high-field regime is related to the localized-magnon physics.

- I noticed some small things which have to be checked.

Page 6/19, I guess, "As shown in Fig. 3(g), the increase of" has to be replaced by "As shown in Fig. 3(f), the increase of" and "as shown in Fig. 3(h). In" by "as shown in Fig. 3(g). In".

Page 7/19, The authors say: "As shown in Fig. 4(a), streak-like excitation at at $Q = 0.8 \text{ \AA}^{-1}$... at 5 K." However, in the figure caption they say: "INS spectra at 4 K". Moreover, I cannot see the streak-like excitation at $Q = 0.8 \text{ \AA}^{-1}$ in Fig. 4(a).

Page 13/19, reference 33, the title is missed.

Page 18/19, figure 4(e), I guess, the label for the x axis should be "E (meV)" instead of "Intensity (a.u.)".

Reviewer #2 (Remarks to the Author):

This manuscript presents an experimental study of a newly-synthesised quantum spin liquid candidate based on the mineral atlasovite, in which magnetic Cu^{2+} ions occupy a potentially frustrated lattice known as square-kagome. The authors show - via a combination of bulk magnetic, muon-spin relaxation, and inelastic neutron measurements - that this material has net antiferromagnetic interactions and does not exhibit long-range magnetic ordering down to ~ 60 mK. They also claim that the magnetic excitation spectrum indicates gapless, spinon-like, excitations.

I think this manuscript describes a significant experimental advance in the field of frustrated magnetism. To the best of my knowledge, there are no other experimental realisations of candidate QSLs on this magnetic lattice, although it has been studied theoretically. The manuscript presents a comprehensive experimental study using a range of experimental techniques. The paper also contains quantum-theoretical modelling, although the link between theory and experiment is somewhat incomplete. Overall, the paper is clearly written and I think can be suitable for Nature Communications.

However, there are several important aspects of the manuscript that, in my opinion, would require improvement before the paper can be accepted for publication. I summarise these below:

(1) The authors make several comments about the absence of structural disorder in this material. However, this assertion is only supported by a qualitative chemical argument: " Furthermore, the Cu^{2+} ions and nonmagnetic ions have different valence numbers [...] avoiding site mixing..", and seems potentially inconsistent with their susceptibility measurements that show "impurity" or "paramagnetic" spins. In my view, more analysis is required to quantify the amount of structural disorder in this material, especially since it has not been reported before. First, the authors should attempt to refine site-disordered models against their synchrotron X-ray data. Second, the authors should give the values of the atomic displacement parameters obtained from their Rietveld refinement. These can often indicate the presence of positional disorder, if they are unusually large.

(2) As part of their basic characterisation, I think the authors should present magnetic susceptibility data collected under both field-cooled and zero-field cooled protocols. Any difference between these two measurements would normally be indicative of (partial) spin freezing. It is important to rule this out in a QSL candidate, and I think that using several complementary techniques here would make the manuscript more convincing.

(3) The authors state: "These INS data are consistent with a gapless continuum of spinon excitations." (p.7)

However, a neutron experiment can only place an upper bound on the size of any gap, because of the finite energy resolution of the instrument. To provide quantitative information, the authors should quantify the maximum gap consistent with their measurements, e.g. by performing fits to an appropriate function (e.g. damped harmonic oscillator) including the effects of instrumental resolution.

(4) In the INS data shown in Fig. 4, I found it surprising that the phonon modes at high Q seem to occur at the same energies as the magnetic modes at low Q . Can the authors comment on this? For example, is it possible that the magnetic excitations are strongly coupled to the lattice? Could the authors perform DFT calculations of the phonon scattering, to determine more accurately the components of the scattering that are purely magnetic?

(5) The authors state "We succeeded in reproducing the magnetic susceptibility and magnetization curve with the J1-J2-J3 SKL model with $J_1 = 135\text{K}$, $J_2 = 162\text{K}$, $J_3 = 115\text{K}$ and $g=2.11\dots$ "

The agreement of this interaction model with the bulk susceptibility data is indeed convincing. However, the authors do not give uncertainties on these exchange interactions, or discuss the procedure they used to estimate them. I am also surprised that bulk measurements contain enough information to determine three exchange parameters accurately. I think the authors should perform further calculations to identify the region(s) of parameter space consistent with their experimental data, and hence provide meaningful confidence intervals on the parameter values they report.

(6) The authors show the calculated Q-dependence of the magnetic scattering intensity from their model in Fig. 5c, and claim it is inconsistent with their experimental data. However, it is hard to tell the extent of this disagreement, because the calculation is of single-crystal scattering whereas the data are powder averaged. I think it would be more informative to powder-average the calculation and integrate it over $0.5 < E < 1.5$ meV, so that it can be compared directly to the experimental data shown in Fig. 4g.

(7) The first sentence of the concluding paragraph states: "We have synthesized a SKL spin-1/2 antiferromagnet [... with] a negligibly weak interlayer coupling." (p.8)
This is a strong statement and I am not convinced it is justified by the evidence shown here. The authors note that their model excluding inter-layer coupling does not account for the experimental INS data, which suggests that further interactions are actually important. The authors did not investigate the effect of including interlayer coupling in their model, so the claim that it is "negligible" seems unproven. I think further calculations are needed here.

(8) The authors claim: "The INS spectrum exhibits a streak-like gapless excitation and flat dispersionless excitation, consistent with powder-averaged spinon excitations." (p.7)
It is not clear to me how a flat dispersionless excitation is consistent with spinon excitations, which are actually strongly dispersive in 1D magnets (see e.g. Nature Physics 9, 435–441 (2013)). The authors should justify this claim with references and/or calculations.

Some more minor points:

(A) I think it should be clarified near the start of the manuscript that all experiments were performed on powder samples. (This is just to prevent possible confusion - I do not see it as a problem.)

(B) "As the released magnetic entropy around T^* is only 8 % of the expected total entropy, this peak cannot be explained by local singlet-triplet excitations in the VBC state." (p.5)
The authors should state, or reference, what entropy release would they expect for a singlet-triplet excitation - otherwise it is not clear why 8% is inconsistent with this scenario.

(C) "The excitation persists up to at least $T = 30$ K (see Fig. S3), which confirms the low-dimensional nature of the magnetic scattering." (p.7)
The persistence of structured magnetic excitations over a wide temperature range is actually seen in many frustrated magnets, including 3D examples such as antiferromagnetic pyrochlores. I therefore think this sentence should therefore be rephrased.

(D) The manuscript contains several typos; a few examples are:

p.2 "excitatos"  "excitations"

p.5 "the the"  "the"

p.5 "Currie-Wiess"  Curie-Weiss

This is not an exhaustive list, and the manuscript would require checking by authors and editors if it is accepted.

Reviewer #3 (Remarks to the Author):

The authors report the first experimental results of the quantum spin liquid (QSL) state on a square-kagame lattice (SKL) quantum antiferromagnet

$\text{KCu}_6\text{AlBiO}_4(\text{SO}_4)_5\text{Cl}$. The evidences from magnetic susceptibility, magnetization, heat capacity, muon spin relaxation, and inelastic neutron scattering are presented and reveal the formation of a gapless QSL at very low temperatures. As a new material, these results are very interesting. However, I have some comments:

The authors claimed that the Cu^{2+} ions and nonmagnetic ions have different valence numbers in this SKL, so that the site mixing can be avoided. It is better to show the experimental evidence that there is no disorder in this SKL, because this is very important for the study on spin liquid. Looking at the INS spectra, the not very clear continuum can also be induced by the effect of disorder.

It is very strange that the ZF-uSR spectra measured at 5K using a DR is different than that in ^4He cryostat. I don't think the authors have fully understand it. In addition, the author didn't pay attention to the fact that the total initial A_{xy} measured in DR is much smaller than that is measured in ^4He cryostat. Is it possible that the sample changed after being measured in one cryostat? These issues should be addressed.

In Fig. 2 (c), the temperature dependence of the total specific heat measured at zero field is shown. However, the nonmagnetic contribution was assumed to include a $\propto T$ linear term, what is that contribution? No labels for the inset of Fig. 2 (c).

To conclude, I think the material and the experimental results are interesting. However, with the those not clear issues I mentioned above, this manuscript does not qualify the standard of Nature Communications.

Reply to Reviewer #1,

Comment (1):

First of all, they must mention somehow that theoreticians studied this model at least since 2002, i.e., before the paper by I. Rousochatzakis et al. (2013) [19]. I mean the following important papers: R. Siddharthan and A. Georges, Square kagome quantum antiferromagnet and the eight-vertex model, Phys. Rev. B 65, 014417 (2002) and J. Richter, J. Schulenburg, P. Tomczak, and D. Schmalfuß, The Heisenberg antiferromagnet on the square-kagome lattice, Condensed Matter Physics 12, 507 (2009); unfortunately, both papers are missed. By the way, in the latter paper one can find the ground-state magnetization curve as it follows from ED ($N = 24 \dots 54$) for the $J_1 = J_2 = J_3$ case (Fig. 5), which looks similar to what is shown in Fig. 5(b) for slightly different values of J_1 , J_2 , and J_3 ($J_2 = 1.2J_1$, $J_3 = 0.85J_1$). Furthermore, the case $J_2 \neq J_3$ was studied in Phys. Rev. B 88, 094426 (2013). The high-field regime is related to the localized-magnon physics.

Answer to Comment (1):

We have added additional references and associated sentences to address these comments.

Comment (2):

- Page 6/19, I guess, “As shown in Fig. 3(g), the increase of” has to be replaced by “As shown in Fig. 3(f), the increase of” and “as shown in Fig. 3(h). In” by “as shown in Fig. 3(g). In”.
- Page 7/19, The authors say: “As shown in Fig. 4(a), streak-like excitation at $Q = 0.8^\circ A^{-1}$... at 5 K.” However, in the figure caption they say: “INS spectra at 4 K”. Moreover, I cannot see the streak-like excitation at $Q = 0.8^\circ A^{-1}$ in Fig. 4(a).
- Page 13/19, reference 33, the title is missed.
- Page 18/19, figure 4(e), I guess, the label for the x axis should be “E (meV)” instead of “Intensity (a.u.)”.

Answer to Comment (1):

Thank you very much for pointing out several mistakes in the original manuscript. According to your comments, we revised the text and figure.

Reply to Reviewer #2,

Comment (1):

The authors make several comments about the absence of structural disorder in this material. However, this assertion is only supported by a qualitative chemical argument: “Furthermore, the Cu^{2+} ions and nonmagnetic ions have different valence numbers [...] avoiding site mixing.”, and seems potentially inconsistent with their susceptibility measurements that show “impurity” or “paramagnetic” spins. In my view, more analysis is required to quantify the amount of structural disorder in this material, especially since it has not been reported before. First, the authors should attempt to refine site-disordered models against their

synchrotron X-ray data. Second, the authors should give the values of the atomic displacement parameters obtained from their Rietveld refinement. These can often indicate the presence of positional disorder, if they are unusually large.

Answer to Comment (1):

We have repeated the crystal structure refinement for the no site mixing and fixed isotropic atomic displacement parameters models. The refinement result is improved, and the low reliability factors and goodness-fit-indicator indicate a satisfactory refinement for $\text{KCu}_6\text{AlBiO}_4(\text{SO}_4)_5\text{Cl}$.

The refinements for the Cu-site-disordered models failed, so we attempted to refine the Cu defect model (the occupancies of atoms were fixed to 1.0, excluding that of Cu). If the Cu is replaced by Al, Bi, K, the occupancy of Cu should be deviated from 1.0. The occupancies of Cu1 and Cu2 sites are refined to 1.004(8) and 0.965(12), indicating the Cu sites have no defect in $\text{KCu}_6\text{AlBiO}_4(\text{SO}_4)_5\text{Cl}$.

Next, we tried to refine the atomic displacement parameters. This refinement result is improved slightly, however, the values converged to an inappropriate value. There are fourteen atomic site in this compound, and six of them are different oxygen sites. In this measurement, we employed a powder sample which was only ground. Thus, there is a possibility that the nonuniformity in the particle diameter exists in the powder sample. Such an effect has a large influence on the refinement of the atomic displacement parameters. Therefore, it is hard to refine the atomic displacement parameters to be the appropriate values.

From the discussion above, we conclude that the Cu^{2+} sites have no defect in $\text{KCu}_6\text{AlBiO}_4(\text{SO}_4)_5\text{Cl}$. We have modified the manuscript to reflect this and have included further data in the supplementary information to describe these findings.

Comment (2):

As part of their basic characterisation, I think the authors should present magnetic susceptibility data collected under both field-cooled and zero-field cooled protocols. Any difference between these two measurements would normally be indicative of (partial) spin freezing. It is important to rule this out in a QSL candidate, and I think that using several complementary techniques here would make the manuscript more convincing.

Answer to Comment (2):

Thank you very much for your indication. We have added the magnetic susceptibility data collected under both field-cooled and zero-field cooled protocols to the Supplementary Information. As shown in Fig. S2, a ZFC/FC divergence is not observed. In addition, the M - H curves show no hysteresis at 1.8 K and 20 K. These results indicate the absence of spin glass behavior, thus suggesting that this compound with an ideal 2D J_1 - J_2 - J_3 SKL has no site-mixing magnetic defect.

Comment (3):

The authors state: "These INS data are consistent with a gapless continuum of spinon excitations." (p.7) However, a neutron experiment can only place an upper bound on the size of any gap, because of the finite energy resolution of the instrument. To provide quantitative information, the authors should quantify the maximum gap consistent with their measurements, e.g. by performing fits to an appropriate function (e.g. damped harmonic oscillator) including the effects of instrumental resolution.

Answer to Comment (3):

You are absolutely right. We examined the fitting that you suggested, but hesitated because the E -dependence of the intensity increases continuously without signature of energy gap at least within the instrumental resolution.

We would like to show the instrumental resolution instead of the upper limit of the gap.

Before revising: Streak-like excitation at $Q = 0.8 \text{ \AA}^{-1}$ is observed at least down to 0.2 meV, and its intensity increases continuously with no energy gap.

After revising: Streak-like excitation at $Q = 0.8 \text{ \AA}^{-1}$ is clearly visible down to the elastic line, and its intensity increases continuously without signature of energy gap at least within the instrumental resolution (FWHM = 0.05 meV for $E_i = 1.69 \text{ meV}$).

Comment (4):

In the INS data shown in Fig. 4, I found it surprising that the phonon modes at high Q seem to occur at the same energies as the magnetic modes at low Q . Can the authors comment on this? For example, is it possible that the magnetic excitations are strongly coupled to the lattice? Could the authors perform DFT calculations of the phonon scattering, to determine more accurately the components of the scattering that are purely magnetic?

Answer to Comment (4):

Thank you very much for your indication. Please find revised figures 4 (b), (c), and S4. Fig. 4(b): Energy dependence of the scattering integrated over Q in the range $1.9 \text{ \AA}^{-1} < Q < 2.1 \text{ \AA}^{-1}$ and $3.9 \text{ \AA}^{-1} < Q < 4.1 \text{ \AA}^{-1}$ measured at 5 K (HRC). Fig. 4(c): Q -dependence of the integrated intensity for the different Gaussian components ($E = 10.1(1) \text{ meV}$, $9.4(3) \text{ meV}$, and $7.3(1) \text{ meV}$), Fig. S4: Energy dependence of the scattering integrated over several ranges in Q measured at 5 K (HRC). The E -dependence of the INS intensity can be fitted well by two or three Gaussian functions and linear baseline, and the corresponding integrated intensities are obtained. As shown in Fig. 4(b), the peak positions of excitations are estimated to be $10.1(1) \text{ meV}$, $9.4(3) \text{ meV}$, and $7.3(1) \text{ meV}$, respectively. The signal due to magnetic excitation is generally enhanced at low- Q values, whereas phonon excitation is dominant at high- Q . As shown in Fig. 4(c), the baseline increase with increasing with Q . Therefore, the baseline may well comes from a number of phonon excitations in a multi-element material $\text{KCu}_6\text{AlBiO}_4(\text{SO}_4)_5\text{Cl}$. The peak at 9.4 meV also increases with increasing with Q , indicating that it comes from phonon excitation. The flat signal at 10.1 meV can be regarded as the magnetic excitations, because the signal due to magnetic excitation is generally enhanced at low- Q values. On the other hand, it is impossible to conclude that the flat signal at around 7.3 meV comes from magnetic excitation, however, we consider it is magnetic mode because its integrated intensity remains almost constant at low- Q . Further study (e.g., the inelastic neutron scattering experiment using the triple-axis spectrometer) is needed to judge whether this signal comes from magnetic excitation because overlapping of streak magnetic excitations. In either case, it's not the case the magnetic mode is developed at the same energies as the phonon modes which we are not interested in this study.

Comment (5):

The authors state "We succeeded in reproducing the magnetic susceptibility and magnetization curve with the J1-J2-J3 SKL model with $J_1 = 135\text{K}$, $J_2 = 162\text{K}$, $J_3 = 115\text{K}$ and $g=2.11\dots$ " The agreement of this interaction model with the bulk susceptibility data is indeed

convincing. However, the authors do not give uncertainties on these exchange interactions, or discuss the procedure they used to estimate them. I am also surprised that bulk measurements contain enough information to determine three exchange parameters accurately. I think the authors should perform further calculations to identify the region(s) of parameter space consistent with their experimental data, and hence provide meaningful confidence intervals on the parameter values they report.

Answer to Comment (5):

We performed additional calculations for the magnetic susceptibility to reply your comments. We verified the validity for our determined exchange interactions in Supplementary Information Sec. V. We compared the calculated results at various ratios with respect to J_1 , J_2 and J_3 with the experimental one. J_{av} defined as $(J_1 + J_2 + J_3) / 3$ can be determined to be 137 K to match the calculated results of the magnetic susceptibility at high temperature region with the experimental one. We can see that the result at $J_1 : J_2 : J_3 = 1 : 0.85 : 1.25$ corresponding to our determined parameters is most consistent with the experimental result, and if the ratio changes even a little, it does not match the experimental one (see Fig. S5). Furthermore, many previous studies have revealed that the g value of the Cu^{2+} ion in the powder samples is about 2.15, which confirms the validity of our result ($g = 2.11$). However, we cannot reproduce the results of neutron scattering with J_1 - J_2 - J_3 SKL, as described in the main text. Therefore, we consider that further investigation of the validity of our parameters is unnecessary. We really hope that you agree with us.

Comment (6):

The authors show the calculated Q -dependence of the magnetic scattering intensity from their model in Fig. 5c, and claim it is inconsistent with their experimental data. However, it is hard to tell the extent of this disagreement, because the calculation is of single-crystal scattering whereas the data are powder averaged. I think it would be more informative to powder-average the calculation and integrate it over $0.5 < E < 1.5$ meV, so that it can be compared directly to the experimental data shown in Fig. 4g.

Answer to Comment (6):

According to your indication, we have revised the Fig. 5(c) and its caption.

Comment (7):

The first sentence of the concluding paragraph states: "We have synthesized a SKL spin-1/2 antiferromagnet [... with] a negligibly weak interlayer coupling." (p.8) This is a strong statement and I am not convinced it is justified by the evidence shown here. The authors note that their model excluding inter-layer coupling does not account for the experimental INS data, which suggests that further interactions are actually important. The authors did not investigate the effect of including interlayer coupling in their model, so the claim that it is "negligible" seems unproven. I think further calculations are needed here.

Answer to Comment (7):

We agree with your comments. Since, the theoretical study of spin state in the SKL spin-1/2 Heisenberg antiferromagnet with interlayer coupling is extremely difficult, we have removed the claim that this compound has a negligibly weak interlayer coupling.

Comment (8):

The authors claim: "The INS spectrum exhibits a streak-like gapless excitation and flat

dispersionless excitation, consistent with powder-averaged spinon excitations." (p.7) It is not clear to me how a flat dispersionless excitation is consistent with spinon excitations, which are actually strongly dispersive in 1D magnets (see e.g. Nature Physics 9, 435–441 (2013)). The authors should justify this claim with references and/or calculations.

Answer to Comment (8):

We have added an additional reference [32] (M. Fujihala, *et al.* Possible Tomonaga-Luttinger spin liquid state in the spin-1/2 inequilateral diamond-chain compound $K_3Cu_3AlO_2(SO_4)_4$. *Sci. Rep.* 7, 16785 (2017)). In $K_3Cu_3AlO_2(SO_4)_4$, spinon continuum is observed in the inelastic neutron scattering spectra, which is in excellent agreement with a theoretical prediction. Please see the figure 2 in ref [32], both simulated and experimental spectra exhibited the flat signal at 10 meV, indicating that there is the van Hove singularity of spinon continuum edges at this energy.

Comment (A):

I think it should be clarified near the start of the manuscript that all experiments were performed on powder samples. (This is just to prevent possible confusion - I do not see it as a problem.)

Answer to Comment (A):

We agree with your suggestion And have added an additional sentence to reflect this.

Comment (B):

As the released magnetic entropy around T^* is only 8 % of the expected total entropy, this peak cannot be explained by local singlet-triplet excitations in the VBC state." (p.5) The authors should state, or reference, what entropy release would they expect for a singlet-triplet excitation - otherwise it is not clear why 8% is inconsistent with this scenario.

Answer to Comment (B):

We agree on your indication and have deleted the corresponding sentences.

Comment (C):

"The excitation persist up to at least $T = 30$ K (see Fig. S3), which confirms the low-dimensional nature of the magnetic scattering." (p.7)

The persistence of structured magnetic excitations over a wide temperature range is actually seen in many frustrated magnets, including 3D examples such as antiferromagnetic pyrochlores. I therefore think this sentence should therefore be rephrased.

Answer to Comment (C):

We would like to revise a sentence to be related to this matter.

Before revising: The excitation persist up to at least $T = 30$ K (see Fig. S3), which confirms the low-dimensional nature of the magnetic scattering.

After revising: The excitation persist up to at least $T = 30$ K (see Fig. S4), which is consistent with the exchange constants estimated later.

Comment (D):

The manuscript contains several typos; a few examples are:

p.2 "excitatio"  "excitations"

p.5 "the the"  "the"

p.5 "Currie-Wiess"  Curie-Weiss

This is not an exhaustive list, and the manuscript would require checking by authors and editors if it is accepted.

Answer to Comment (D):

Thank you very much for pointing out typos. We have now corrected them.

Reply to Reviewer #3,

Comment (1):

The authors claimed that the Cu²⁺ ions and nonmagnetic ions have different valence numbers in this SKL, so that the site mixing can be avoided. It is better to show the experimental evidence that there is no disorder in this SKL, because this is very important for the study on spin liquid. Looking at the INS spectra, the not very clear continuum can also be induced by the effect of disorder.

Answer to Comment (1):

Please see a revised Supplemental material and Answer to Comment (1) and (2) of Reviewer #2. The result of the Rietveld refinement strongly suggests that Cu sites have no defect. In addition, magnetic susceptibilities measured under ZFC and FC conditions indicates the absence of spin glass behavior are indicated. We consider that these are the experimental evidence that there is no disorder in this SKL.

Comment (2):

It is very strange that the ZF-uSR spectra measured at 5K using a DR is different than that in ⁴He cryostat. I don't think the authors have fully understand it. In addition, the author didn't pay attention to the fact that the total initial Asy measured in DR is much smaller than that is measured in ⁴He cryostat. Is it possible that the sample changed after being measured in one cryostat? These issues should be addressed.

Answer to Comment (2):

We have carried out both μ SR measurements using different pellets which are made from powder sample of same synthesis batch. Therefore, we consider that the measurements were performed using same quality of samples.

For capable of uniformly cooling the pellet, we prepared a smaller pellet than that used in ⁴He cryostat, accordingly changed the diameter of collimator. It is the cause of why the total initial asymmetries measured in DR is smaller than that is measured in ⁴He cryostat.

We could not specify what is the cause of extrinsic component. However, the cause of that may be in refrigerator because both measurements were performed using same quality of samples. In fact, the intrinsic component is the majority of the total asymmetry, suggesting that the pellet had not deteriorated.

Comment (3):

In Fig. 2 (c), the temperature dependence of the total specific heat measured at zero field is shown. However, the nonmagnetic contribution was assumed to include a T linear term, what is that contribution? No labels for the inset of Fig. 2 (c).

Answer to Comment (3):

A T linear term does not have a physical significance, therefore, we deleted it and recalculated the entropy. The released magnetic entropy is slightly increased.

Reviewers' comments:

Reviewer #1 (Remarks to the Author):

Gapless spin liquid in a square-kagome lattice antiferromagnet

by

M.Fujihala et al.

In the first-round referee's reports on the submitted original manuscript, there were several interesting comments and questions requiring further studies and changes of the paper.

In particular,

the authors were asked

to provide further evidence for the absence of structural disorder,

to rule out the spin glass behavior,

to rethink again about the INS data and to redo theoretical DMRG calculations for the scattering intensity,

to rephrase statements in several places of the manuscript,

to modify the list of references,

as well as

to remove typos.

I think, the authors have provided satisfactory answers to all referee's comments and the revised version of the paper may be published in Nature Communications.

However, there are still some small mistakes

(e.g.,

p.6/21: Supplementary Information Sec. II -> Supplementary Information Sec. III;

p.8/21: finite temperature lanczos -> finite temperature Lanczos;

p.11/21: luttinger liquid -> Luttinger liquid;

p.13/21: references 19 and 21, the list of authors must be checked;

p.19/21: Supplementary Information Sec. II -> Supplementary Information Sec. III;

p.2/5, Fig.S1: figure capture looks unfinished

etc.).

Reviewer #2 (Remarks to the Author):

I would like to thank the authors for their careful and detailed responses to my questions. They have answered all the points that I raised, and I am happy to recommend this manuscript for publication in Nature Communications.

There is just one point that I would like the authors to check before the paper is finally accepted. The authors note that their model [Fig. 5c] appears inconsistent with their neutron-scattering data, because the model shows a main peak at $Q = 1.3 \text{ \AA}^{-1}$ whereas the data show the main peak at $Q = 0.8 \text{ \AA}^{-1}$. I would like the authors to double-check that the x-axis units of Fig. 5c are correct: are they definitely $Q (\text{ \AA}^{-1})$ and not the reciprocal-lattice units (dimensionless) favoured by theorists? A confusion between these units would cause the x-axis of the calculation to be stretched by a factor $a/(2\pi) = 1.6$, which would exactly explain the apparent discrepancy with the experimental data. (I apologise if the authors already checked this!)

Reviewer #3 (Remarks to the Author):

I found the comments of Referee 2 to be very important and completely agree with him/her. I do not think the authors have adequately addressed the questions raised, especially about having 2% free-spins but yet having the Cu layers structurally perfect. I think 2% impurity spins can screw up everything. This is also my concern 1 in my previous report.

For the issue in MuSR experiment, I do not think the authors have understood it completely. Why the initial total asymmetry is smaller measured in DR. They claim that the reason is that they used a smaller pellet in DR. This is totally wrong! The initial asymmetry does not depend on the size of sample. If the sample is smaller, then the signal from the background will be larger, so that the dashed line in Figure S3 would be higher. The total asymmetry should not be changed.

For the specific heat measurements, now in Figure 2 (c), the lattice contribution is much smaller than the measured one even at high temperatures. Previously, they have a $\langle T \rangle$ linear term in the lattice contribution. I do not think the $\langle T \rangle$ linear can be simply deleted. One needs to know where it is from and understand it. In addition, the entropy calculated is much smaller than $R \ln 2$, it seems that the authors do not understand it as well.

Reply to Reviewer #1,

Comment (1):

- p.6/21: Supplementary Information Sec. II -> Supplementary Information Sec. III.
- p.8/21: finite temperature lanczos -> finite temperature Lanczos.
- p.11/21: luttinger liquid -> Luttinger liquid.
- p.13/21: references 19 and 21, the list of authors must be checked.
- p.19/21: Supplementary Information Sec. II -> Supplementary Information Sec. III.
- p.2/5, Fig.S1: figure capture looks unfinished.

Answer to Comment (1):

Thank you very much for pointing out several mistakes in the original manuscript. According to your comments, we revised the text and figure.

Reply to Reviewer #2,

Comment (1):

There is just one point that I would like the authors to check before the paper is finally accepted. The authors note that their model [Fig. 5c] appears inconsistent with their neutron-scattering data, because the model shows a main peak at $Q = 1.3 \text{ \AA}^{-1}$ whereas the data show the main peak at $Q = 0.8 \text{ \AA}^{-1}$. I would like the authors to double-check that the x-axis units of Fig. 5c are correct: are they definitely $Q \text{ (}\text{\AA}^{-1}\text{)}$ and not the reciprocal-lattice units (dimensionless) favoured by theorists? A confusion between these units would cause the x-axis of the calculation to be stretched by a factor $a/(2\pi) = 1.6$, which would exactly explain the apparent discrepancy with the experimental data.

Answer to Comment (1):

Thank you very much for your indication. We re-confirmed the x-axis units of Fig. 5c. And we are convinced that it is right.

Reply to Reviewer #3,

Comment (1):

I found the comments of Referee 2 to be very important and completely agree with him/her. I do not think the authors have adequately addressed the questions raised, especially about having 2% free-spins but yet having the Cu layers structurally perfect. I think 2% impurity spins can screw up everything. This is also my concern 1 in my previous report.

Answer to Comment (1):

We consider that the 2.4 % component is due to the extrinsic impurity or unpaired spins on surface of powder particles.

The results of XRD and ZFC/FC susceptibility measurements indicate the absence of spin glass behavior.

The magnetisation curve measured at 1.8 K, as shown in the inset of Fig. 2(b), has two components: an intrinsic component M_{bulk} and free spin component M_{free} . Following the

analysis for herbertsmithite, a saturated magnetisation of M_{free} can be estimated by subtraction of the linear M_{bulk} from the measured total magnetisation M_{obs} . The M_{free} can be fitted a Brillouin function for a spin-1/2, suggesting the component is attributed to the paramagnetic impurity or the unpaired spins on surface of powder particles.

Comment (2):

For the issue in MuSR experiment, I do not think the authors have understood it completely. Why the initial total asymmetry is smaller measured in DR. They claim that the reason is that they used a smaller pellet in DR. This is totally wrong! The initial asymmetry does not depend on the size of sample. If the sample is smaller, then the signal from the background will be larger, so that the dashed line in Figure S3 would be higher. The total asymmetry should not be changed.

Answer to Comment (2):

I understand there is no correlation between the pellet sizes and the initial asymmetries. The initial asymmetries will change in the different experimental setups. This is because the conditions near the sample such as the wall of the vacuum vessel of the cryostat, the number of windows, and the thickness of the sample holder change. Please see Fig. 2, page 34 of KEK-MSL REPORT 2012 (<https://lib-extopc.kek.jp/preprints/PDF/2013/1323/1323003.pdf>). This experiment was performed in the same beam line using a same dilution refrigerator, and is test experiment by using blank silver sample holder without sample. The initial total asymmetry (~ 0.18) is similar to that of our results.

The important point is that the relaxation rates match at the same temperature, despite different setups and initial asymmetries.

Comment (3):

For the specific heat measurements, now in Figure 2 (c), the lattice contribution is much smaller than the measured one even at high temperatures. Previously, they have a T linear term in the lattice contribution. I do not think the T linear can be simply deleted. One needs to know where it is from and understand it. In addition, the entropy calculated is much smaller than $R \ln 2$, it seems that the authors do not understand it as well.

Answer to Comment (3):

One common problem in the analysis of the magnetic specific heat for low-dimensional magnets is the estimation of the lattice contribution, especially because magnetic coupling extends up to much higher temperatures. In the present case we also could not get a precise estimation of the lattice contribution because the compound which the Cu^{2+} ions are replaced by the nonmagnetic ions was not obtained. The lattice contribution is intimately connected to the vibration modes of the ions, and in most cases, it cannot be fitted by the Debye model. In previous version, to estimate the released entropy value, we modeled a lattice contribution as the odd function. Thus, a T linear term in the lattice contribution did not have a special meaning. Magnetic entropy obtained by integration of $(C - C_{\text{lattice}})/T$, however, the lattice contribution in the very low temperature region is very small. Therefore, even if there is a deviation in the assumed curve and lattice contribution, the estimated magnetic entropy at around T^* is not so much affected. However, there is a possibility a T linear term may give the reader some misunderstandings, thus we adopted the assumed lattice contribution $C_{\text{lattice}} = 0.000555 T^3$.

The magnetic correlations develop at a much higher temperature than T^* . As shown in the

Fig. 2(a), there is a deviation in the inverse magnetic susceptibility and Curie-Weiss law at around 150 K. Thus the magnetic entropy release should be observed in the high temperature region, however, there is no way for us to determine that. Thus it is impossible to identify the ground state of $\text{KCu}_6\text{AlBiO}_4(\text{SO}_4)_5\text{Cl}$ from the specific heat measurement. In the calculated specific heat of the spin-1/2 kagome Heisenberg antiferromagnet (KHAF), small peak appear at around $T = J/100$, and the released entropy at around the peak has estimated about 20 % of total entropy [J. Schnack et al PRB 98, 094423 (2018)]. Similar characteristics are also observed in $\text{KCu}_6\text{AlBiO}_4(\text{SO}_4)_5\text{Cl}$, however, we have no evidence for these are attributable to a same origin with KHAF. Careful consideration is necessary about this issue. We therefore concluded only that the long-range magnetic is not observed.

REVIEWERS' COMMENTS:

Reviewer #4 (Remarks to the Author):

I went through the revised manuscript, the comments from the reviewers and the answers to the comments.

There were several points raised by the referees which have been answered satisfactorily to by the authors, according to the referees but there seemed to be some remaining controversy about the muSR experiment which I address below. At the end of this report, please find a few additional remarks after I read the manuscript.

Indeed, after a first really misleading response by the authors to explain the difference between asymmetries in different cryogenic set ups, the authors have referred to a technical report dated 2013 showing that the measured asymmetry in the dilution refrigerator (DR) is consistent with their data. Yet, this does not explain why the asymmetry is larger in the He4 cryostat, for a given set of detectors such as that from the D1 beamline, usually it goes in the opposite way as the multiplicity of walls in the DR usually leads to a selection of higher energy for the emitted positrons then an increased asymmetry. There are two possibilities for explaining this:

An upgrade occurred in the set-up of D1 instrument, 0.18 is the asymmetry before the upgrade [Journal of Physics: Conference Series 551(2014) 012063] and it was about 0.205 [JPS Conf. Proc. , 011062 (2018) for the S1 spectrometer], anyway < 0.24 after the upgrade of D1 identical to that of the S1 spectrometer. The authors should check with the beamline scientist whether this qualifies for explaining partly the Asymmetry difference, if the experiments in the two setups were performed before and after the upgrade.

An alternative and very likely additional possibility is that the sample mounted in the He4 cryostat was much more massive leading to an increased asymmetry for a given set-up.

Anyway, I believe that such details are not commonly mentioned neither in the main body of manuscripts nor in Supplementary Materials, even in full muSR papers. I agree that the third reviewer's criticisms should have been handled in a proper manner once being raised, his question was clear and definitely calling for a much more expert answer, the least I can say, than the first given one. Overall, the DR experiment per se clearly shows that all the asymmetry is detected from 58 mK up to 5K, hence there is no hidden frozen component and the fully detected signal is purely dynamical at base-T, the major conclusion from the muSR study – unless one parasitic phase would freeze at very high temperatures but it would have been detected through x-rays.

In passing, I have two comments:

(i) references to gapped vs ungapped discussion in herbertsmithite look completely out of date. Better use the more appropriate NMR references: M. Fu, Science, P. Khuntia, Nature Physics

(ii) disqualifying the quantum spin liquid physics in herbertsmithite by advocating the scenario disorder from K. Kawamura Sensei, is not correct, as the amount of requested disorder is certainly not present in herbertsmithite.

(iii) the Curie tail in the susceptibility has not received any explanation other than free spins. Yet attributing to grain surface effect does not make sense in a few % ratio for standard powers, hence this interesting square-kagome lattice antiferromagnet suffers from some defects, a common feature to all these kagome-based materials up to now. This does not obscure at all the fact that the discovery of this material is a nice achievement.

(iv) the fluctuating field seems quite small from the decoupling experiment. This could have been discussed in the paper.

Reply to Reviewer #4,

Comment:

There were several points raised by the referees which have been answered satisfactorily to by the authors, according to the referees but there seemed to be some remaining controversy about the muSR experiment which I address below. At the end of this report, please find a few additional remarks after I read the manuscript.

Indeed, after a first really misleading response by the authors to explain the difference between asymmetries in different cryogenic set ups, the authors have referred to a technical report dated 2013 showing that the measured asymmetry in the dilution refrigerator (DR) is consistent with their data. Yet, this does not explain why the asymmetry is larger in the He4 cryostat, for a given set of detectors such as that from the D1 beamline, usually it goes in the opposite way as the multiplicity of walls in the DR usually leads to a selection of higher energy for the emitted positrons than an increased asymmetry. There are two possibilities for explaining this:

An upgrade occurred in the set-up of D1 instrument, 0.18 is the asymmetry before the upgrade [Journal of Physics: Conference Series 551(2014) 012063] and it was about 0.205 [JPS Conf. Proc. , 011062 (2018) for the S1 spectrometer], anyway < 0.24 after the upgrade of D1 identical to that of the S1 spectrometer. The authors should check with the beamline scientist whether this qualifies for explaining partly the Asymmetry difference, if the experiments in the two setups were performed before and after the upgrade.

An alternative and very likely additional possibility is that the sample mounted in the He4 cryostat was much more massive leading to an increased asymmetry for a given set-up.

Anyhow, I believe that such details are not commonly mentioned neither in the main body of manuscripts nor in Supplementary Materials, even in full muSR papers. I agree that the third reviewer's criticisms should have been handled in a proper manner once being raised, his question was clear and definitely calling for a much more expert answer, the least I can say, than the first given one. Overall, the DR experiment per se clearly shows that all the asymmetry is detected from 58 mK up to 5K, hence there is no hidden frozen component and the fully detected signal is purely dynamical at base-T, the major conclusion from the muSR study – unless one parasitic phase would freeze at very high temperatures but it would have been detected through x-rays.

Answer to Comment:

Thank you very much for your indication. Your comments are exactly right. An upgrade occurred in the set-up of D1 instrument, and the initial asymmetry 0.18 was the asymmetry before the upgrade. When I confirmed the starting date of the experiments, both experiments were performed after the upgrade.

The possibility that the difference between asymmetries in different cryogenic set ups is attributed to the different sample weights is high.

The sample mounted in the ^4He cryostat was much more massive than that mounted in a

dilution refrigerator.

To uniformly cool, we prepared a thin pellet which is much lighter than that mounted in the ^4He cryostat. In addition, the completely different environments in a dilution refrigerator and ^4He cryostat, the wall of the vacuum vessel, number of windows, thickness of the sample holder, etc., may be one of the causes of this difference.

The complex internal structure of a dilution refrigerator probably makes the initial asymmetry change, because the multiple scattering of positrons, dephasing of the muon polarisation before muons reach the sample, etc. may occur.

Comment (i):

(i) references to gapped vs ungapped discussion in herbertsmithite look completely out of date. Better use the more appropriate NMR references: M. Fu, Science, P. Khuntia, Nature Physics

Answer to Comment (i):

According to your suggestions, we changed the references about your comments.

Comment (ii):

disqualifying the quantum spin liquid physics in herbertsmithite by advocating the scenario disorder from K. Kawamura Sensei, is not correct, as the amount of requested disorder is certainly not present in herbertsmithite.

Answer to Comment (ii):

According to your suggestions, we deleted a reference and sentence about valence-bond glass state.

Comment (iii):

the Curie tail in the susceptibility has not received any explanation other than free spins. Yet attributing to grain surface effect does not make sense in a few % ratio for standard powers, hence this interesting square-kagome lattice antiferromagnet suffers from some defects, a common feature to all these kagome-based materials up to now. This does not obscure at all the fact that the discovery of this material is a nice achievement.

Answer to Comment (iii):

We could not extract the intrinsic contribution from $\chi(T)$ because the measurement temperature range is not enough. Therefore, we could not discuss details about this issue. There is a need in further study.

Comment (iv):

the Curie tail in the susceptibility has not received any explanation other than free spins. Yet attributing to grain surface effect does not make sense in a few % ratio for standard powers, hence this interesting square-kagome lattice antiferromagnet suffers from some defects, a common feature to all these kagome-based materials up to now. This does not obscure at all the fact that the discovery of this material is a nice achievement.

Answer to Comment (iv):

We couldn't get at the exact meaning of this comment you gave us. Similar LF dependence of the muon spin relaxation rate was observed in other QSL candidates (e.g. PHYSICAL REVIEW B 94, 024438 (2016)). To understand this, we should prioritize to reveal what kind of QSL state is realized in this compound because the quantum states of each QSL are

substantially different. There is a need in further theoretical and experimental study.